# A Meta-Analysis of Influencing Factors on the Activity of BiVO_4_-Based Photocatalysts

**DOI:** 10.3390/nano13162352

**Published:** 2023-08-16

**Authors:** Ruijie Che, Yining Zhu, Biyang Tu, Jiahe Miao, Zhongtian Dong, Mengdi Liu, Yupeng Wang, Jining Li, Shuoping Chen, Fenghe Wang

**Affiliations:** 1School of Environment, Nanjing Normal University, Nanjing 210023, China; 2120210253@glut.edu.cn (R.C.); 222502023@njnu.edu.cn (Y.Z.); 212501004@njnu.edu.cn (B.T.); jh963266@dal.ca (J.M.); 222512031@njnu.edu.cn (M.L.); 2School of Materials Science and Engineering, Guilin University of Technology, Guilin 541010, China; 3Key Laboratory for Soft Chemistry and Functional Materials of Ministry of Education, Nanjing University of Science and Technology, Nanjing 210094, China; dztnoone@foxmail.com; 4School of Pharmacy, Nanjing Technology University, Nanjing 211816, China; 202261109033@njtech.edu.cn

**Keywords:** meta-analysis, BiVO_4_-based composites, degradation efficiency, BET, kinetic constant

## Abstract

With the continuous advancement of global industrialization, a large amount of organic and inorganic pollutants have been discharged into the environment, which is essential for human survival. Consequently, the issue of water environment pollution has become increasingly severe. Photocatalytic technology is widely used to degrade water pollutants due to its strong oxidizing performance and non-polluting characteristics, and BiVO_4_-based photocatalysts are one of the ideal raw materials for photocatalytic reactions. However, a comprehensive global analysis of the factors influencing the photocatalytic performance of BiVO_4_-based photocatalysts is currently lacking. Here, we performed a meta-analysis to investigate the differences in specific surface area, kinetic constants, and the pollutant degradation performance of BiVO_4_-based photocatalysts under different preparation and degradation conditions. It was found that under the loading condition, all the performances of the photocatalysts can be attributed to the single BiVO_4_ photocatalyst. Moreover, loading could lead to an increase in the specific surface area of the material, thereby providing more adsorption sites for photocatalysis and ultimately enhancing the photocatalytic performance. Overall, the construct heterojunction and loaded nanomaterials exhibit a superior performance for BiVO_4_-based photocatalysts with 136.4% and 90.1% improvement, respectively. Additionally, within a certain range, the photocatalytic performance increases with the reaction time and temperature.

## 1. Introduction

Due to the growing population, the rise in global industrialization, and the over-consumption of natural resources, environmental pollution problems are becoming increasingly serious, of which water pollution is one of the major global environmental problems [1]. China is one of the world’s largest producers and consumers of pesticides, with an annual use of more than 20,000 tons [2]. Moreover, agricultural production heavily relies on chemical fertilizers [3]. Large amounts of organic pollutants from pesticides and fertilizers mix with agricultural runoff under rainwater, exacerbating the pollution of water bodies [4]. Additionally, Mining activities and the industrial production of paper, textile, leather, iron and steel, and petroleum discharge a large amount of printing and dyeing wastewater. Electroplating and metallurgical wastewater contain a large amount of organic pollutants [5,6], including detergents, organic dyes, surfactants, curing agents, etc. [7]. Furthermore, inorganic pollutants such as Cr(VI) are also contained in wastewater released from industry [8]. These pollutions not only lead to the serious pollution of the water habitat, but also pose certain risks to organisms and human health [6].

At present, the main means to remediate water pollution are chemical precipitation, ion exchange, redox, adsorption, photocatalysis, and biotechnology [9,10,11]. Among these technologies, photocatalysis is the best choice for degrading organic and inorganic pollutants in the environment. The principle of photocatalysis is to convert solar energy into chemical energy to produce a catalytic reaction and to excite the surrounding oxygen atoms and water molecules to form highly oxidized free radicals [12]. Photocatalysis enables the efficient use of solar energy and the degradation of most organic and inorganic pollutants that are harmful to the environment and humans [13]. Compared with other pollutant removal methods, photocatalytic technology has the advantages of low cost, high efficiency, environmental friendliness, and sustainability [14,15,16].

Photocatalysts facilitate oxidation and reduction reactions in the presence of light energy [17]. Excellent photocatalysts are important for the realization of photocatalytic technology. Prominent examples include g-C_3_N_4_-based photocatalysts [18], TiO_2_-based photocatalysts [19], Bi-based photocatalysts [20,21], Nb_2_O_5_-based photocatalysts [22], etc. Among them, TiO_2_, ZnO, and other common photocatalysts are not suitable for large-scale commercial applications in practice due to the absorption of UV light, fast electron–hole pair complexation, and a wide bandgap [23,24]. Compared with the above photocatalysts, Bi-based photocatalysts are a class of visible-light-driven high-efficiency photocatalysts with wider application prospects [25]. Among the Bi-based photocatalysts, BiVO_4_ possesses outstanding superiority in terms of preparation cost as well as energy band width, e.g., a narrower band gap of 2.4 eV compared with TiO_2_, ZnO, and a superior valence band position, good corrosion resistance, good dispersion, and high photocatalytic stability. In addition, BiVO_4_ is abundant in the Earth’s crust, which also makes it eligible for large-scale applications [26,27,28,29,30]. However, BiVO_4_ photoanodes typically suffer from severe photo-corrosion and low charge carrier mobility (~4 × 10^−2^ cm^2^ V^−1^ s^−1^), which leads to electron–hole complexation, with most of the photogenerated carriers disappearing before reaching the photoanode–electrolyte interface. Also, it has a short hole diffusion distance (~70 nm), the photocurrent density of the pure BiVO_4_ catalyst is still very low compared to the theoretical limit of 7.5 mA cm^−2^ (100 mW cm^−2^), and the performance is far from the theoretical optimum. Thus, it is necessary to find and develop multiple ways to overcome the above drawbacks and improve the photocatalytic activity of BiVO_4_ [31,32,33,34,35]. There are many means to enhance the photocatalytic activity of BiVO_4_, including changing the morphology, self-doping, and coupling metal or semiconductor [36,37,38]. Among the various methods, elemental doping can effectively modulate the electrical properties of materials and is one of the effective means to improve photocatalytic activity [39,40,41]. For example, the photocatalytic efficiency of BiVO_4_/WO_3_ nanocomposites prepared by hydrothermal method was as high as 92.02% [42]. Z-type heterojunction composites of NiFe_2_O_4_/{010}BiVO_4_ synthesized by photo-deposition technique can remove environmental pollutant NOR up to 98.97% efficiently [43]. The doping of different ratios of g-C_3_N_4_ can significantly enhance the removal efficiency of pollutants [44].

A comparative analysis of the results across studies using a global dataset provides a more intuitive understanding of how elemental doping affects the degradation of environmental pollutants by BiVO_4_ photocatalysts. Therefore, in this study, the effect of BiVO_4_-based composites on the removal of environmental pollutants was investigated using meta-analysis. Based on our research, no existing meta-analysis or systematic review has explored on the effectiveness of BiVO_4_-based composites on the removal of organic and inorganic pollutants from the environment. In this study, we collected the relevant literature to examine the influence of various preparation and degradation conditions on the BET and kinetic constants of BiVO_4_-based composites. Subsequently, the effect of BiVO_4_-based composites on the removal of environmental pollutants, including organic as well as inorganic pollutants, under different conditions has been investigated. Our results can provide guidance and assistance for the practical application of BiVO_4_-based composites in the field of environmental pollution remediation.

## 2. Literature Extraction and Data Screening

### 2.1. Literature Search

In this paper, the peer-reviewed papers so far were searched using Web of Science and ScienceDirect databases. The search was conducted with “BiVO_4_” OR “‘bismuth vanadium tetraoxide’ OR ‘bismuth vanadate‘” AND “Photocatalysis” AND “degradation” AND “‘contaminants’ OR ‘Dyes’ OR ‘Antibiotics’ OR ‘Surfactant’ OR ‘Pesticides’” OR “‘BiVO_4_’ OR ‘bismuth vanadium tetraoxide’ OR ‘bismuth vanadate’” AND “Photocatalysis” AND “reduction” AND “Heavy Metal” searched as keywords to compare the pollutant degradation and removal efficiency of single BiVO_4_ as a photocatalyst and composite photocatalytic material. The initial search resulted in 322 papers. Since the beginning of research in this field in 2005, the number of articles published and the frequency of citations in this field are shown in Figure 1. Subsequently, a secondary screening of the selected literature was further conducted based on the following factors.

### 2.2. Criteria for Data Inclusion

This study included articles in the meta-analysis based on the following criteria: (1) original, peer-reviewed articles published in English; (2) all studies compared the efficiency of single BiVO_4_ photocatalysts with composite photocatalyst materials for the degradation of pollutants; and (3) studies that were not meta-analyses, conference proceedings, reports, etc., to prevent the duplication of data. After applying the above criteria and excluding duplicate articles, 228 papers were screened from the initial database. Subsequently, 187 articles were excluded based on the relevance of their titles and keywords. Afterward, we excluded seven of these articles after reviewing the selected articles due to missing or quality issues of the main data of the meta-analysis, etc. Finally, 34 most relevant articles were selected for data extraction and analysis after a rigorous screening review process (Figure 2).

### 2.3. Data Extraction and Grouping

The data extracted from each article include the preparation conditions of the material (coupling type, preparation time, and reaction temperature) and the degradation conditions (contaminant type, contaminant concentration, photocatalyst dosing, and solution pH). Additionally, the specific surface area, degradation efficiency, and cycling performance of the materials in the single BiVO_4_ photocatalyst (control) and the composite photocatalyst coupled with other materials (treatment group), as well as their mean and standard deviation, were also counted. The BiVO_4_-based composites obtained under different preparation methods have different morphologies, properties and advantages (Table 1).

The types of coupling in the preparation conditions are divided into five categories: (1) metals (mainly Zn, Ag, etc.) [45,46,47]; (2) oxides (ZnO, MnO_2_, TiO_2_, Cu_2_O, etc.) [48,49,50,51]; (3) sulfides (ZnIn_2_S_4_, CuS, etc.) [52,53]; (4) carbon materials; (5) nanomaterials; and (6) heterojunctions. Preparation times are classified as 1–6 h, 6–24 h, and more than 24 h. Reaction temperatures are classified as low temperature (below 100 °C), medium temperature (between 100 and 200 °C), and high temperature (above 200 °C).

The degradation conditions encompass both organic and inorganic pollutants. The concentration of pollutants is divided into three categories: (1) <20 mg/L; (2) 20–200 mg/L; (3) >200 mg/L. The amount of photocatalysts added is also divided into three categories: (1) <0.5 g/L; (2) 0.5–1 g/L; (3) >1 g/L. Solution pH is divided into three categories: acidic, neutral, and alkaline.

## 3. Meta Analysis

### 3.1. Selection of Models

Two types of models used in the Meta-analysis are fixed effects models and random effects models. In the fixed effects model, each independent study is from the same overall sample, while in the random effects model, the studies are from different overall samples [78]. The random effects model was chosen as the statistical model for analysis in this study as it more accurately captures the variation in true effect sizes [79]. In this study, samples were taken from photocatalytic degradation studies of BiVO_4_ under different experimental designs, and the normal effect sizes of the selected samples logically had different properties, such as the methodological background of the study, the loaded compounds, and the conditions of degradation, all with different effect values. Therefore, the random effects model is more applicable to this study.

### 3.2. Calculation of the Effect Size

In this study, the effects of material preparation conditions and degradation conditions on BiVO_4_ single photocatalysts and BiVO_4_ coupled with composite photocatalysts of other materials were investigated using the effect size method. The standard measure of effect size is expressed as the logarithmic response ratio (*lnRR*). The logarithmic response ratio was calculated as follows [80]:(1)$$lnRR=ln\frac{W_{t}}{W_{c}}=lnW_{t}-lnW_{c}$$
where *W_t_* and *W_c_* are the mean values of each test variable in the experimental and control groups, respectively.

A positive effect size (*lnRR* > 0) indicates an increase in the tested variables (degradation efficiency, BET, and kinetic constants) of the composite photocatalyst made of BiVO_4_ coupled with other materials compared to the control BiVO_4_ single photocatalyst, while a negative effect size (*lnRR* < 0) indicates a decrease in the tested variables of the experimental group compared to the control group. An effect size of 0 (*lnRR* = 0) indicates that there is no significant change in the experimental group compared to the control group. The percentage of the magnitude of the effect size was calculated according to the following formula [48]:(2)$$Effect\;size\;\left(\%\right)=\left(e^{\ln RR}-1\right)\times 100$$

The variance of the effect size is calculated by the following equation [81]:(3)$$v=\frac{{{SD}_{t}}^{2}}{n_{t}{W_{t}}^{2}}+\frac{{{SD}_{c}}^{2}}{n_{c}{W_{c}}^{2}}$$
where, *SD_t_*, and *SD_c_* represent the standard deviation of each test variable in the experimental and control groups, respectively, and *n_t_* and *n_c_* represent the number of repetitions in the experiments of the experimental and control groups, respectively. The weighted response ratio to (*RR_++_*) is used to express the overall response effect of variable conditions such as reaction time on the experimental and control groups, which is calculated as follows [82]:(4)$${RR}_{++}=\frac{\sum_{i-1}^{n}\sum_{j-1}^{k}\frac{1}{v}{RR}_{ij}}{\sum_{i-1}^{n}\sum_{j-1}^{k}\frac{1}{v}}$$
where *n* denotes the number of experimental groups, the weighting factor is represented by the inverse of the variance of the effect size 1/*v*, *i* and *j* represent the *i*-th and *k*-th treatments, respectively, and *k* is the observed value in the observed group.

The standard deviation of *RR_++_*(*s*) is:(5)$$s=\sqrt{\frac{1}{\sum_{i-1}^{n}\sum_{j-1}^{k}\frac{1}{v}}}$$

The 95% confidence interval (*CI*) in the study was calculated according to the following formula [83]:(6)$$95\%CI={RR}_{++}\pm 1.96\times s({RR}_{++})$$

### 3.3. Subgroup Analysis

In this study, the material preparation conditions as well as the degradation conditions listed in Section 2.3 were used as subgroups for a subgroup analysis of each test variable in the experimental as well as the control groups. Because this study used a randomized model for analysis, the selected trials were measured inconsistently and in different units of measure, so standardized mean difference (SMD) was used to express the results of each test variable between the experimental and control groups [84]. *SMD* is a relative indicator and is independent of baseline risk and has good consistency. The effect size in this study is reported as *SMD*. The indicator was calculated by the following equation:(7)$$SMD=\frac{MD}{SD}$$
where, *MD* and *SD* denote the mean and standard deviation of each test variable in the experimental and control groups, respectively.

### 3.4. Data Analysis

The meta-analysis in this study was performed by applying the meta package in R language (4.2.3) together with Review Manager 5.3 software. The final result plots were plotted using the ggplot2 package in the R language (4.2.3).

## 4. Results and Discussion

### 4.1. Effects of Different Preparation Conditions on BET of BiVO_4_-Based Composites

The specific surface area of the composites is one of the main factors affecting their photocatalytic activity; therefore, understanding the specific surface area of photocatalytic materials is essential to grasp the performance of photocatalysts. A higher specific surface area provides more active sites [44], which in turn increases the photocatalytic activity of the material [85]. The current method of measuring the specific surface area and pore volume of photocatalytic materials mainly uses the nitrogen adsorption Brunauer–Emmett–Teller (BET) method [86]. In this study, the effect of BET on the BiVO_4_-based composites was counted in the collected 34 related papers (Figure 3 and Appendix A). In the listed preparation conditions, for coupling type, the effect size of BET is maximized when composited with nanomaterials (12.23, 95% CI = 11.65, 12.80), followed by loaded oxides (6.22, 95% CI = 5.80, 6.64) and heterojunctions (5.98, 95% CI = 4.46, 7.51). In this study, the coupling-type loadings with the highest removal efficiency for all pollutants were screened (Appendix A). Meanwhile, for the preparation time, the effect size of the BET method of BiVO_4_-based composites gradually decreases with increasing preparation time. Specifically, the highest effect size is observed for a preparation time of 1–6 h (9.8, 95% CI = 9.1, 10.49), followed by preparation for 6–24 h (3.45, 95% CI = 3.2, 3.7) and preparation for more than 24 h (0.67, 95% CI = −0.04, 1.37). The preparation temperature also has different effects on the rate of the BET method effect of BiVO_4_-based composites. The effect size of the BET of the composites was greatest when the preparation temperature was in the mid-temperature range of 100–200 °C (5.87, 95% CI = 5.56, 6.18), while the effect size of the BET of the composites was lowest at low temperature conditions (<100 °C) (1.88, 95% CI = 1.50, 2.25). At the same time, the pH of the solution during the preparation process is one of the main factors affecting the BET of BiVO_4_-based composites. According to the results of the meta-analysis, the effect size of the BET of BiVO_4_-based composites decreases with increasing pH, with the highest effect size under acidic conditions (12.06, 95% CI = 11.40, 12.73) and the lowest under alkaline conditions (3.30, 95% CI = 2.50, 4.10), indicating that the preparation in acidic environment has a greater potential for enhancing the specific surface area of the composites.

All the preparation conditions listed in this study (including material coupling type, material preparation time, material preparation temperature, and pH at preparation) have a positive effect on the specific surface area effect of BiVO_4_-based composites, indicating that BiVO_4_-based composites have a larger specific surface area than single materials regardless of the preparation conditions, and the composites can provide more adsorption and active sites for photocatalytic degradation and reduction in environmental pollutants.

### 4.2. Effects of Different Preparation Conditions on Kinetic Constant of BiVO_4_-Based Composites

The kinetic constant is a parameter that describes the rate of reaction and the concentration between reactants, representing the rate of degradation per unit time of the reactant concentration. A higher kinetic constant value indicates that the photocatalytic material has a faster rate and higher efficiency in the degradation of or reduction in the pollutant [87]. The effects of the preparation conditions and degradation conditions selected in this study on the kinetic constants are shown in Figure 4. For the preparation conditions, the highest effect sizes of kinetic constants among the coupling types were found for carbon materials (2.4, 95% CI = 0.48, 4.32), followed by oxides (0.58, 95% CI = −0.1, 1.26), sulfides (0.43, 95% CI = −2.89, 3.75), and nanomaterials (0.26, 95% CI = −0.94, 1.47); the effect size of the kinetic constants of metal couples was negative for all coupling types (−0.81, 95% CI = −1.46, −0.15), indicating that the coupling of composites prepared from heavy metals may lead to a decrease in the kinetic constants of the composites. There was little difference in the effect of preparation time on the kinetic constant effect size, which increased with preparation time, with the highest kinetic constant effect size for preparation time >24 h (0.83, 95% CI = −1.46, 3.12). In addition, the composite kinetic constants had the highest effect sizes when the preparation temperature was in the range of 100–200 °C (0.76, 95% CI = 0.06, 1.45) and the lowest effect sizes when the preparation temperature was above 200 °C (0.36, 95% CI = −0.98, 1.7). Finally, the preparation pH also influenced the kinetic constants of the composites, with the highest effect sizes of kinetic constants for pH-neutral conditions (1.71, 95% CI = −0.02, 3.44), while there was minimal variation in the effect sizes of kinetic constants for acidic and basic conditions (Figure 4 and Appendix A).

The effect size of the reaction kinetic constants was also affected by the different pollutant concentrations in the degradation conditions. The effect size of the reaction kinetics of BiVO_4_-based composites was highest when the contaminant concentration was 20–200 mg/L (0.97, 95% CI = −1.17, 3.11), followed by the case of contaminant concentration < 20 mg/L (0.46, 95% CI = −0.45, 1.37), whereas when the contaminant concentration was higher than 200 mg/L, the kinetic constants of effect was the lowest (0.05, 95% CI = −0.90, 1.01), almost coinciding with the baseline, indicating that the composite material at this concentration has little effect on the kinetic constants. The amount of photocatalyst dosing also has an effect on the rate of kinetic constant effect of the composite. Specifically, the highest effect size of the kinetic constants of the composites was observed when 0.5–1 g/L of photocatalyst was injected (0.77, 95% CI = 0.23, 1.32), the investigation has demonstrated that the precise selection of the optimal photocatalyst dosage holds the key to maximizing the rate of photocatalytic reactions. However, both an excessive and an inadequate dosage of photocatalyst proved to be insufficient in achieving the desired outcome. Specifically, a lower dosage results in a diminished number of available sites for photocatalytic reactions, whereas an elevated photocatalyst concentration diminishes the solution’s light transmittance, consequently exerting a notable influence on the reaction rate. In addition, for the solution pH at degradation, the highest effect sizes of kinetic constants for composites were found for pH-neutral conditions (0.46, 95% CI = −0.28, 1.21), while the effect sizes for acidic and basic conditions did not differ significantly, which is consistent with the results of the effect of pH on the effect sizes of kinetic constants in the preparation conditions.

### 4.3. Study on the Degradation Efficiency of BiVO_4_-Based Composites for Environmental Pollutants

Environmental pollutants, including organic and inorganic pollutants, pose a serious threat to the natural environment and human health [88]. BiVO_4_ and BiVO_4_-based composites, which are non-toxic to the environment, can effectively degrade various organic and inorganic pollutants in water bodies [89,90]. In this study, the efficiency of BiVO_4_-based composites in degrading environmental pollutants under various conditions was investigated in order to better understand the photocatalytic performance of various composites. The degradation efficiency of BiVO_4_-based composites for environmental pollutants is affected by a combination of the material preparation conditions (including coupling type, preparation time, preparation temperature, and pH) and degradation conditions (pollutant concentration, photocatalyst dosing, and solution pH during degradation). The effect sizes of the degradation efficiency of BiVO_4_-based composites under each condition are shown in Figure 3. Regardless of the preparation and degradation conditions, the degradation rate of BiVO_4_-based composites was improved compared to that of single BiVO_4_ material (Figure 5 and Table 2).

Among the preparative conditions, the highest effect sizes for the degradation efficiency of BiVO_4_-based composites were found when heterojunctions were constructed (13.93, 95% CI = 12.73, 15.13), followed by coupling nanomaterials (13.76, 95% CI = 12.96, 14.56). On the other hand, the lowest effect size of composite degradation efficiency was observed for all types of coupling conditions when coupling oxides (4.33, 95% CI = 3.94, 4.73). The effect size of the degradation efficiency of BiVO_4_-based composites increased with longer preparation time; and the highest effect size of degradation efficiency of composites was observed when the preparation time exceeded 24 h (15.19, 95% CI = 14.26, 16.12), while the lowest effect size of degradation efficiency was observed for a degradation time of 1–6 h (6.96, 95% CI = 6.72, 7.20). Similarly, the effect size of the degradation efficiency of the composites increased with a higher preparation temperature. The highest effect size of degradation efficiency (11.97, 95% CI = 11.31, 12.62) was observed for high temperature preparation conditions (>200 °C) and the lowest effect size of degradation efficiency (4.89, 95% CI = 4.65, 5.12) was observed for low temperature preparation conditions (<100 °C). In addition, the pH at the time of preparation also had an effect on the effect size of the composite degradation efficiency, which was the highest when the pH was 7, i.e., under neutral conditions (11.64, 95% CI = 10.92, 12.36), followed by basic (10.92, 95% CI = 10.33, 11.52) and acidic conditions (8.87, 95% CI = 8.33, 9.41). For the degradation conditions, different pollutant concentrations have varying effects on the effect sizes. The highest effect sizes for degradation efficiency were found for pollutant concentrations of 20–200 mg/L (11.18, 95% CI = 10.66, 11.69), which coincided with the results of the reaction kinetic constants. The lowest effect size of composite degradation efficiency was observed when the pollutant was at a low concentration (<20 mg/L) (7.08, 95% CI = 6.84, 7.33). The photocatalyst dosage emerged as a pivotal factor within the degradation conditions. The analysis revealed a discernible trend wherein the magnitude of the degradation efficiency’s effect size for BiVO4-based composites grew proportionately with the photocatalyst dosage within a specific range. Notably, the effect size reached its zenith at a dosage of 0.5–1 g/L (7.26, 95% CI = 7.00, 7.52), resulting in an incremental degradation efficiency of 112.02% relative to that of the solitary BiVO_4_ material. This augmentation can be attributed to the enhanced provision of reactive sites brought forth by a greater photocatalytic concentration, propelling the photocatalytic reaction. However, when the dosage exceeded 1 g/L, the contribution of loading to the photocatalytic performance of the composites was reduced (4.88, 95% CI = 4.56, 5.21). This attenuation can be attributed to higher concentrations, leading to a reduction in solution transmittance, thereby adversely affecting the photocatalytic performance. The solution pH during degradation also influenced the effect size of the composite degradation efficiency, and similar to the results of the reaction kinetic constants, the highest effect size of the degradation rate was observed for neutral pH conditions (14.81, 95% CI = 13.79, 15.82), followed by acidic (9.47, 95% CI = 9.02, 9.91) and alkaline conditions (4.6, 95% CI = 4.02, 5.19).

The above results show that coupling is a highly effective method to improve the degradation or reduction in environmental pollutants by BiVO_4_ photocatalysts, and the various coupling materials prepared by different methods exhibit diverse morphological characteristics and degradation advantages (Table 1). The photocatalytic degradation and reduction in environmental pollutants are influenced and controlled by several direct and indirect factors, such as the type of material coupling, preparation pH, preparation time, preparation temperature, as well as the concentration of pollutants in the degradation conditions, photocatalyst dosage, and solution pH.

#### 4.3.1. Efficiency of Degradation of Organic Pollutants

The types and amounts of organic pollutants in the water environment are increasing due to the growing pollution caused by industrial, agricultural, and household production activities, etc. [91]; for example, dyeing and textile wastewater, organic pesticide wastewater and industrial wastewater from food, pharmaceuticals, electroplating, metallurgy, etc., which contain significant amounts of organic pollutants [92]. Photocatalysis is currently a low-cost, efficient, and environmentally friendly means of degrading organic pollutants in water bodies [26], among the 34 relevant literatures screened in this study, as many as 32 focused on the degradation of organic pollutants in the environment. Consistent with the analysis of the overall degradation efficiency, the BiVO_4_-based composites outperformed the unloaded single BiVO_4_ photocatalysts under various preparation conditions as well as degradation conditions (Figure 6 and Appendix A).

The highest effect size of BiVO_4_-based composites on the degradation efficiency of organic pollutants (14.29, 95% CI = 13.33, 15.25) was observed when coupling nanomaterials in the preparation conditions, because the porous structure of nanomaterials can greatly increase the specific surface area of the composites, thus improving the photocatalytic performance of BiVO_4_-based composites [93]. This was followed by construct heterojunctions (11.92, 95% CI = 10.86, 12.97) as well as metals (11.52, 95% CI = 10.67, 12.34). The effect size of the composite degradation efficiency of organic pollutants increased with longer material preparation time, and the highest effect size of composite degradation efficiency was observed when the preparation time was greater than 24 h (14.61, 95% CI = 13.71, 15.5). Similarly, the effect size of the degradation efficiency of BiVO_4_-based composites increased with a higher preparation temperature, with the highest effect size of the composite degradation efficiency when the preparation temperature was higher than 200 °C (13.01, 95% CI = 12.30, 13.72). It is shown that the longer preparation time and high temperature conditions have the highest degree of optimization for the degradation efficiency of the composites to degrade organic pollutants within a certain range. The effect of solution pH at preparation on the effect size of composite degradation efficiency was consistent with the overall results, with the highest effect size of composite degradation efficiency when the pH was neutral (13.01, 95% CI = 11.83, 14.19), followed by alkaline (10.92, 95% CI = 10.33, 11.52) and acidic conditions (8.87, 95% CI = 8.33, 9.41). It is shown that the BiVO_4_-based composites have a better degradation efficiency for environmental organic pollutants than single photocatalysts when the pH at the time of preparation is around 7.

For degradation conditions, the highest effect sizes for composite degradation efficiency were found for pollutant concentrations of 20–200 mg/L (15.65, 95% CI = 14.94, 16.35), followed by concentrations above 200 mg/L (13.89, 95% CI = 12.55, 15.23) and below 20 mg/L (10.04, 95% CI = 9.67, 10.40). It was shown that the prepared composites degraded environmental organic pollutants in the medium concentration range with more positive excursions and a better degradation efficiency than the uncoupled single photocatalyst. The findings of this study illuminate a noteworthy trend, wherein the most pronounced effect size pertaining to the enhancement of the degradation efficiency of environmental organic pollutants (7.50, 95% CI = 7.23, 7.76) is discernible within the photocatalyst dosage range of 0.5–1 g/L. Evidently, augmenting the photocatalyst quantity within a certain range amplifies the photocatalytic prowess of the composites to a greater degree. However, an excessive photocatalyst dosage tends to mitigate the extent of enhancement in the photocatalytic efficiency of the composites. The excessive dosage appears to be associated with a decreased solution transmittance, which consequently engenders a reduction in photocatalytic performance. This observation resonates harmoniously with the outcomes of earlier analyses. In addition, the pH of the solution during the degradation of environmental organic pollutants had an effect on the rate of effect on degradation efficiency, and calculations showed that acidic (13.31, 95% CI = 12.70, 13.93) and neutral (12.74, 95% CI = 11.86, 13.62) conditions contributed more to the degradation efficiency of organic pollutants. In contrast, under alkaline degradation conditions (3.41, 95% CI = 3.11, 3.71), although the effect size increased, the increment was smaller compared to acidic and neutral conditions.

#### 4.3.2. Efficiency of Degradation of Inorganic Pollutants

Agricultural and industrial activities emit a large amount of wastewater containing heavy metal pollutants and greenhouse gas emissions into the environment [94]. In addition, photocatalytic means can effectively remove inorganic pollutants such as NO, NO_x_, and CO from air pollutants [95]. In contrast to the extensive research on the photocatalytic removal of environmental organic pollutants, there are fewer studies on inorganic pollutant removal. After conducting a thorough screening process, only two relevant papers on the removal of environmental inorganic pollutants by BiVO_4_-based composites were identified. The meta-analysis of these selected studies revealed that the removal efficiency of inorganic pollutants (mainly the reduction in heavy metals in wastewater) increased after the coupling of a single BiVO_4_ material compared to the case without coupling (Appendix A).

For the preparation conditions of the composites, the main types of coupling used for the reduction in inorganic pollutants in the environment were nanostructures and heterojunctions, where the BiVO_4_-based composites showed a higher rate of reduction efficiency effect on inorganic pollutants after loading nanomaterials (10.16, 95% CI = 8.96, 11.36) and a better coupling effect. For the preparation time of the selected composites in the literature, similar to the effect size results for the overall degradation efficiency, the longer the preparation time, the larger the positive offset in the effect size for the inorganic pollutant reduction efficiency, and for the reduction in inorganic pollutants, the effect size was higher for 6–24 h of preparation (11.11, 95% CI = 9.81, 12.42). Similarly, as the preparation temperature increases, the rate of the effect of the composite on the reduction efficiency of environmental inorganic pollutants rises. The highest effect size of reduction efficiency was observed when the preparation temperature was higher than 100 °C (11.69, 95% CI = 10.32, 13.06), as shown in the analytical results. For the solution pH at preparation, the effect size increment on the removal efficiency of environmental pollutants was lower for the removal of inorganic pollutants (9.04, 95% CI = 8.27, 9.81) than for the removal of organic pollutants under neutral conditions with higher effect size increments, indicating that the BiVO_4_-based composites are more efficient in degrading organic pollutants under the same conditions, but there are fewer original data on the reduction in inorganic pollutants, and so the conclusion may not be generalizable.

For each reduction condition of the composites, similar to the above analysis, the photocatalyst dosing was proportional to the effect size of the inorganic pollutant reduction efficiency, with the highest effect size of the reduction efficiency of the BiVO_4_-based composites seen for the reduction in environmental inorganic pollutants when the dosing was 0.5–1 g/L (5.63, 95% CI = 5.11, 6.16), which is consistent with the above findings. Regardless of the pollutant concentration and the pH of the solution adjusted for the reduction in inorganic pollutants, the results of the analysis showed that the reduction in environmental inorganic pollutants was enhanced by the loading of BiVO_4_ materials, with a greater increase in the effect of the efficiency of the reduction in inorganic pollutants under alkaline conditions compared to the degradation of organic pollutants (8.12, 95% CI = 7.14, 9.10). Due to the small amount of raw data, further analysis will not be performed here.

### 4.4. Multiple Factors Jointly Determine the Photocatalytic Activity of BiVO_4_-Based Photocatalysts

Through a diligent review of the meticulously selected literature and an exhaustive meta-analysis, we have acquired an initial grasp of the efficacy exhibited by BiVO_4_-based composites in eliminating environmental pollutants across varying conditions. The interplay of factors such as coupling type, preparation duration, reaction temperature, and degradation pH collectively contributes to shaping the ultimate degradation performance of these composites. Notably, all of the aforementioned conditions exert a beneficial influence on elevating the removal efficiency of environmental pollutants.

#### 4.4.1. Effect of Coupling Type on Photocatalytic Activity of Composites

For the coupling type, the rate of effect on pollutant removal efficiency can be maximized when BiVO_4_-based heterojunctions are constructed successfully (13.93, 95% CI = 12.73, 15.13), followed by coupled nanomaterials (13.76, 95% CI = 12.96, 14.56) and sulfides (8.68, 95% CI = 8.00, 9.36) (Figure 5 and Table 2). By examining the screened literature, it was found that when heterojunctions are successfully constructed, the pollutant removal efficiency of the composites can be better improved. The formation of heterojunctions can increase the specific surface area of the composites and effectively narrow the band gap width of BiVO_4_, which can accommodate a larger wavelength range of visible light [49]. In addition, the formation of heterojunctions can inhibit the compounding of photogenerated electron–hole pairs by promoting the transfer of photogenerated carriers, which improves the degradation efficiency of BiVO_4_-based composites to a certain extent [96]. In the realm of photocatalytic reactions, heterojunctions are commonly classified into three primary types: type II heterojunctions, Z-type heterojunctions, and S-type heterojunctions. In the context of a type II heterojunction, the semiconductors’ band gaps are ingeniously interleaved, engendering disparate valence band energies that prompt charge carriers to migrate in opposite directions, thereby enhancing charge separation effectiveness [97]. Conversely, the Z-type heterojunction’s surface features a modest polarization effect, leading to the robust trapping of photogenerated electron pairs, consequently impeding their recombination. Remarkably, the Z-type heterojunction showcases a heightened redox activity, effectively prolonging the migration cycle of photogenerated electron pairs. This temporal extension ultimately bolsters the efficiency of photocatalytic reactions by elongating the lifespan of photogenerated carriers [98,99]. On another front, the S-type heterojunction exerts adept control over carrier transfer pathways, selectively retaining valuable electrons within the conduction band (CB). This strategic curation facilitates the proficient separation of electrons, culminating in an elevated photocatalytic performance [100]. The porous structure of the nanomaterials enables a strong interaction between BiVO_4_ and the nanomaterials, which enhances the separation of photogenerated carriers [56], leading to the enhancement of the degradation properties of the composites. Sulfide coupling can inhibit carrier binding to a certain extent and accelerate electron–hole separation, leading to an increase in photocurrent density and thus enhancing the photocatalytic performance of BiVO_4_-based composites [101].

Relatively speaking, the positive shifts in the photocatalytic efficiency effect sizes of BiVO_4_-based composites coupled with metals (8.09, 95% CI = 7.49, 8.69), oxides (4.33, 95% CI = 3.94, 4.73), and carbon materials (7.49, 95% CI = 7.02, 7.96) are somewhat smaller (Table 2 and Figure 5). The amalgamation of plasma metal nanoparticles with semiconductor photocatalytic materials introduces lattice distortion and assumes the role of electron traps on the semiconductor, thereby fostering electron–hole pair separation and interfacial electron transfer, ultimately heightening photocatalytic efficacy [102]. However, the photocatalytic performance of the composites is easily affected by the structure of noble metal ions such as Au, Ag, and their positions on the semiconductor photocatalysts, which makes the photocatalytic performance under metal-loaded conditions less stable [103]. The oxide-based composites primarily revolve around semiconductor combinations, exemplified by BiVO_4_-integrated oxides, which enable the efficient segregation of electrons and holes in opposite directions. Concurrently, the synergy of semiconductors with varying bandgap widths extends the light absorption spectrum of the composites, bolstering photocatalytic efficiency [104]. However, as a trade-off, coupling with narrow-bandwidth semiconductors often engenders challenges concerning stability, cost, and preparatory complexity. Similarly, the incorporation of carbon materials contributes to an enhanced photocatalytic efficacy by adeptly suppressing electron–hole recombination. Nonetheless, these composites also contend with stability issues [105].

#### 4.4.2. Effect of Preparation Time and Preparation Temperature on Photocatalytic Activity of Composites

The preparation time of the composites also has an effect on the effect size of the degradation efficiency. In the literature selected for this study, the preparation time was categorized into three fractions, 1–6 h, 6–24 h, and more than 24 h, and the effect size of the degradation efficiency of the composites increased with longer preparation time of the materials, with the effect size reaching its maximum at more than 24 h of preparation (15.19, 95% CI = 14.26, 16.12) (Figure 5 and Table 2). A longer preparation time provides more contact time for BiVO_4_-based composites for photocatalytic performance enhancement [105]. The composite preparation temperatures were categorized into three types according to the literature selected for this study, low temperature (≤100 °C), medium temperature (100–200 °C), and high temperature (≥200 °C), in which the effect size of the pollutant degradation efficiency of the composites was the largest when high temperature preparation was used (11.97, 95% CI = 11.31, 12.62), while that of the pollutant degradation efficiency for low temperature preparation was the lowest (4.89, 95% CI = 4.65, 5.12) (Figure 5 and Table 2). It is shown that the amount of effect on the degradation efficiency of environmental pollutants of BiVO_4_-based composites increases with the temperature of material preparation within a certain range. A higher preparation temperature allows BiVO_4_ to obtain better-ordered mesoporous conditions, and the crystallinity of BiVO_4_ increases with increasing temperature, thus increasing the rate of the effect of the composites in degrading pollutants [106]. The effect of preparation conditions on the specific surface area of the composites also showed that preparation at medium and high temperatures had a better promotion for the increment of the specific surface area effector of BiVO_4_-based composites (Figure 3), and this results in an increase in the specific surface area of the composite, which in turn increases the degradability of the composite.

#### 4.4.3. Effect of Degradation pH on Photocatalytic Activity of Composites

The pH value of a solution is considered a significant factor affecting photocatalytic activity because it induces changes in the characteristics of both pollutants and photocatalysts. In the case of ciprofloxacin (CIP), when the pH of the solution is above 8.9, the CIP molecules exist in the form of anions. Conversely, when the pH is below 3.6, the CIP molecules exist in the form of cations [107,108]. Furthermore, an excess of H^+^ and OH^−^ ions can react with superoxide anion radicals (O_2_^−^) and/or compete for adsorption, leading to a reduced removal efficiency of CIP in acidic and alkaline solutions. In contrast, within the pH range of 3.6 to 8.9, CIP adopts a neutral form. In this neutral form, the CIP ring possesses a higher electron density, making it more susceptible to photo-degradation by the reactive species generated in the photocatalytic system. Therefore, under neutral conditions, the photocatalyst can more effectively degrade CIP [108].

As delineated by our aforementioned meta-analysis, the most substantial effect size concerning the photocatalytic efficacy of BiVO_4_-based composites in pollutant degradation was observed (11.64, 95% CI = 10.92, 12.36) when operating within neutral pH conditions (refer to Figure 5 and Table 2). Notably, the utilization of BiVO_4_ matrix composites within this neutral milieu resulted in pronounced positive shifts in the effect sizes of photocatalytic performance for both organic (13.01, 95% CI = 11.83, 14.19) and inorganic (9.04, 95% CI = 8.27, 9.81) pollutants within the environmental spectrum (as demonstrated in Figure 6 and detailed in Appendix A).

#### 4.4.4. Other Factors Affecting Photocatalytic Efficiency

Numerous additional factors wield influence over photocatalytic reactions. Among these, the light source, encompassing both duration and intensity, also exerts an impact on photocatalytic efficiency. Within the purview of photocatalytic technology, the array of light sources encompasses sunlight [109], simulated sunlight [110], ultraviolet lamps [111], xenon lamps [112], mercury lamps [109], among others. Notably, roughly half of solar energy constitutes visible light, with a mere 4–6% being near-ultraviolet (UV) light capable of exciting photocatalysts. However, the majority of contemporary photocatalysts exhibit inefficiencies in harnessing sunlight. Consequently, there is an imperative to engineer photocatalysts that display a heightened responsiveness to visible light, a realm in which BiVO_4_ occupies a prominent position as a significant class of visible-light-driven photocatalysts. Wang et al. compared the photocatalytic performance of BiVO_4_-based photocatalysts under UV and visible light by evaluating rhodamine B as a target pollutant and found that C-doped BiVO_4_-based photocatalysts had better photocatalytic activity under visible light irradiation [113]. Zhang et al. (2021) found that the degradation performance of photocatalysts for pollutants increased with the increase in light duration and light intensity over a range of that which is found in [114]. This further indicates that the light source is also one of the possible influencing factors for photocatalyst activity.

The exposed crystalline surface of semiconductor photocatalysts is also one of the important factors affecting photocatalytic efficiency. A good crystalline surface structure can enable the photogenerated electrons and holes of the photocatalysts to realize a more efficient separation, and Srinivasan et al. (2022) found that BiVO_4_-based photocatalysts grown along the (121) and (040) crystalline surfaces had a promotive effect on the photocatalytic performance of the photocatalysts by adjusting the various preparation parameters [115]. Tian (2020) et al. found that BiVO_4_-based photocatalysts with (010) and (110) as the main exposed crystalline surfaces have different photocatalytic performances from those found in [116]. All of the above indicate that the crystal surface exposure is closely related to the catalytic performance of photocatalysts.

In addition, the photocatalytic performance of BiVO_4_-based composites is also closely related to their particle size. When the size of the composite particles is as small as the nanometer scale, a quantum size effect occurs. As the size decreases, the forbidden band width of the semiconductor becomes larger and the redox capacity becomes stronger. At the same time, the smaller the size, the shorter the time for electron–hole pairs to reach the surface of the catalyst and the faster the reaction with the adsorbed material on the surface of the catalyst [117,118]. The synergistic coupling of g-C_3_N_4_ with BiVO_4_ into ultrathin nanosheets has been discerned as a potent enhancer of the photocatalytic performance exhibited by the resulting composites. [119]. A reduced particle size contributes to a heightened photocatalytic reaction performance by amplifying the exposure of active sites [120].

## 5. Conclusions and Outlook

### 5.1. Conclusions

In this study, a systematic evaluation of the performance of BiVO_4_-based composites for the removal of organic and inorganic pollutants from the environment was performed using meta-analysis. The preparation conditions, including coupling type, preparation time, and reaction temperature, as well as the pollutant degradation conditions, including pollutant type, pollutant concentration, photocatalyst dosage, and solution pH, which can affect the pollutant degradation efficiency, were reviewed. This study is the first to analyze the environmental pollution degradation performance of BiVO_4_-based composites using meta-analysis, aiming to find out the optimal conditions for BiVO_4_-based composites to degrade environmental pollutants by photocatalytic technology in practical applications through systematic comparative analysis. BiVO_4_-based composites with different preparation conditions exhibit distinct morphological and performance advantages (Table 1). The results of the above meta-analysis show that BiVO_4_-based composites exhibit a higher removal efficiency for environmental pollutants compared to single BiVO_4_ materials, regardless of whether they are doped with metals, oxides, sulfides, carbon materials, nanomaterials, or construct heterojunctions. Furthermore, other preparation conditions can also improve the performance of the composites under certain conditions. Composites enhance the effectiveness of their removal of environmental organic and inorganic pollutants by increasing the specific surface area of the material and enhancing the reaction kinetic constants.

### 5.2. Outlook and Future Perspective

While numerous studies have documented the pertinent attributes of BiVO_4_-based photocatalysts, there remains a plethora of issues that warrant thorough consideration and resolution in the future:(1)According to the results of our meta-analysis, in order to make the photocatalytic performance of BiVO_4_-based composites as high as possible, the method should be designed to optimize the various preparation and reaction conditions of the materials to construct nanoscale heterojunctions with large specific surface area. And the preparation pH should be controlled to be around neutral, while allowing each preparation material be fully mixed at a higher temperature.(2)BiVO_4_-based photocatalysts have demonstrated a promising capacity for effectively degrading organic pollutants present in wastewater, including substances like rhodamine B and methylene blue. Additionally, these photocatalysts exhibit a notable proficiency in reducing inorganic heavy metals, such as Cr(VI), within aqueous environments. Although current research primarily encompasses laboratory-based simulations and verifications, there exists a substantial opportunity for future endeavors to transition toward large-scale applications in real-world wastewater treatment processes.(3)BiVO_4_-based photocatalysts, responsive to visible light, present a cost-effective alternative to conventional photocatalytic devices, offering greater environmental compatibility. In the material synthesis process, prioritizing the utilization of economically viable and environmentally benign raw materials for coupling with BiVO_4_ helps prevent potential secondary pollution. This strategic approach not only ensures the mitigation of material preparation-related environmental impacts, but also paves the way for feasible large-scale industrial applications.

In conclusion, further research is required to identify BiVO_4_-based photocatalysts with optimal photocatalytic performance under specific preparation conditions as well as degradation conditions and incorporate them into large-scale production applications. This involves the continuous optimization of complex types and their loadings to maximize the photocatalytic performance of the composites and to explore their photocatalytic effects on different types of pollutants. In addition, it is essential to develop preparation methods that are cost-effective, energy-efficient, environmentally friendly, and suitable for industrial applications.

## Figures and Tables

**Figure 1 nanomaterials-13-02352-f001:**
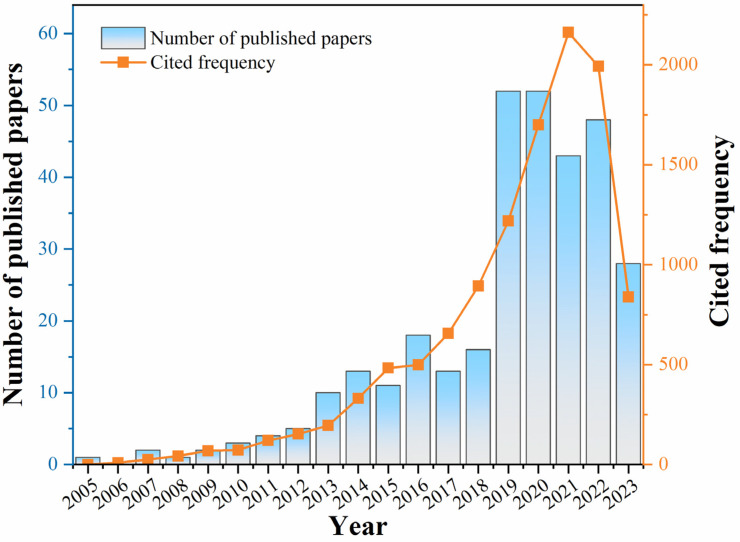
Number of publications and citation frequency of the related literature since 2005.

**Figure 2 nanomaterials-13-02352-f002:**
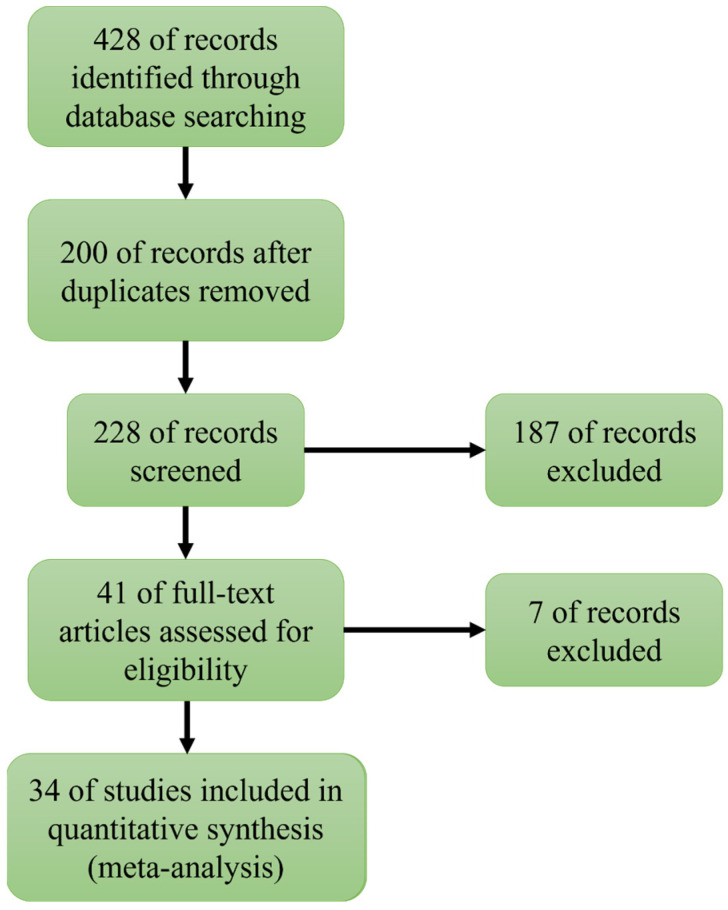
Flow chart of the meta-analysis of the literature inclusion and exclusion.

**Figure 3 nanomaterials-13-02352-f003:**
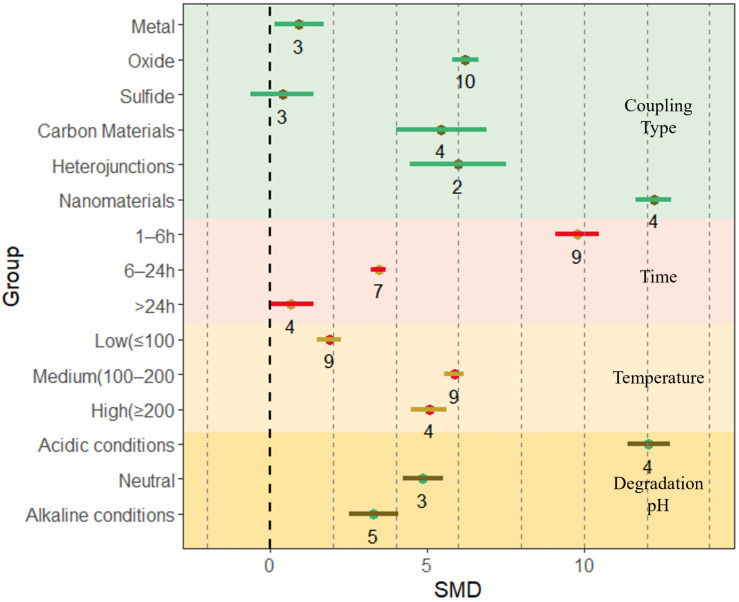
The BET of BiVO_4_-based composites is mainly influenced by the type of coupling, degradation time, degradation temperature, and degradation pH. Horizontal error bars indicate 95% confidence intervals. (Colored lines indicate effect size intervals and numbers indicate the amount of literature for that component).

**Figure 4 nanomaterials-13-02352-f004:**
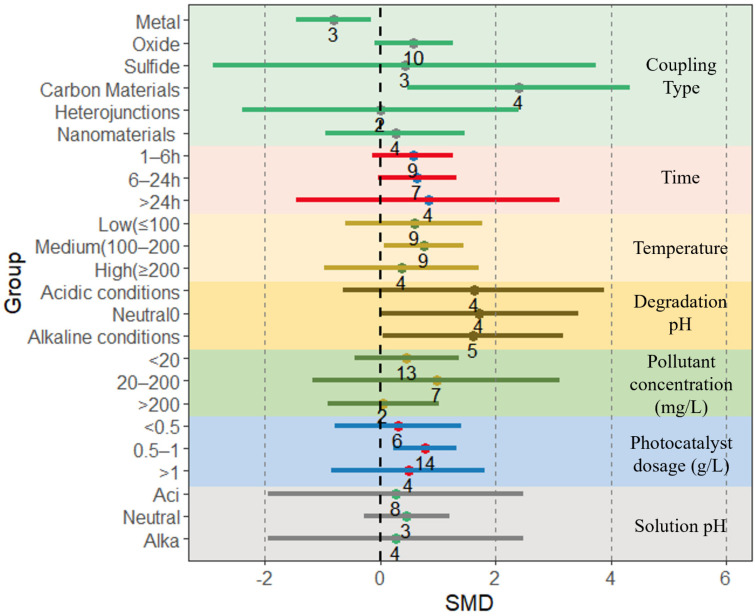
The kinetic constants of BiVO_4_-based composites are mainly influenced by the type of coupling, degradation time, degradation temperature, degradation pH, pollutant concentration, photocatalyst dosage, and solution pH. Horizontal error bars indicate 95% confidence intervals. (Colored lines indicate effect size intervals and numbers indicate the amount of literature for that component).

**Figure 5 nanomaterials-13-02352-f005:**
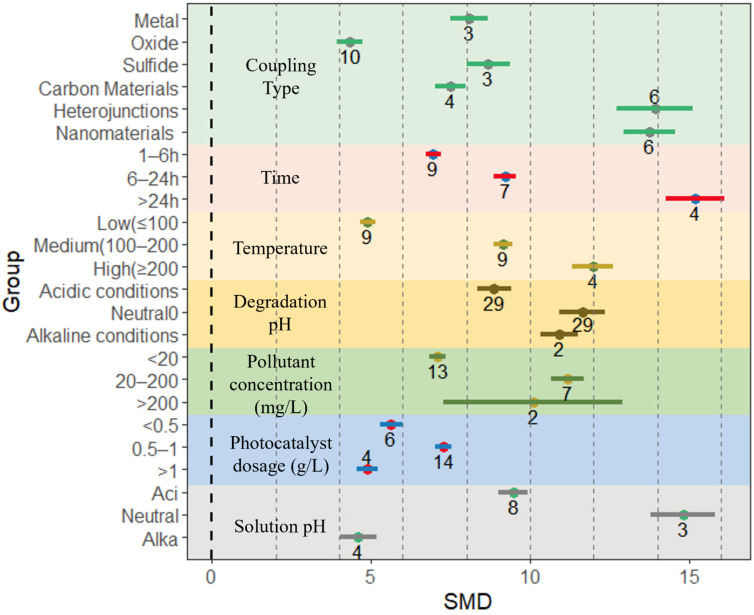
The degradation efficiency of BiVO_4_-based composites is mainly influenced by the type of coupling, degradation time, degradation temperature, degradation pH, pollutant concentration, photocatalyst dosage, and solution pH. Horizontal error bars indicate 95% confidence intervals. (Colored lines indicate effect size intervals and numbers indicate the amount of literature for that component).

**Figure 6 nanomaterials-13-02352-f006:**
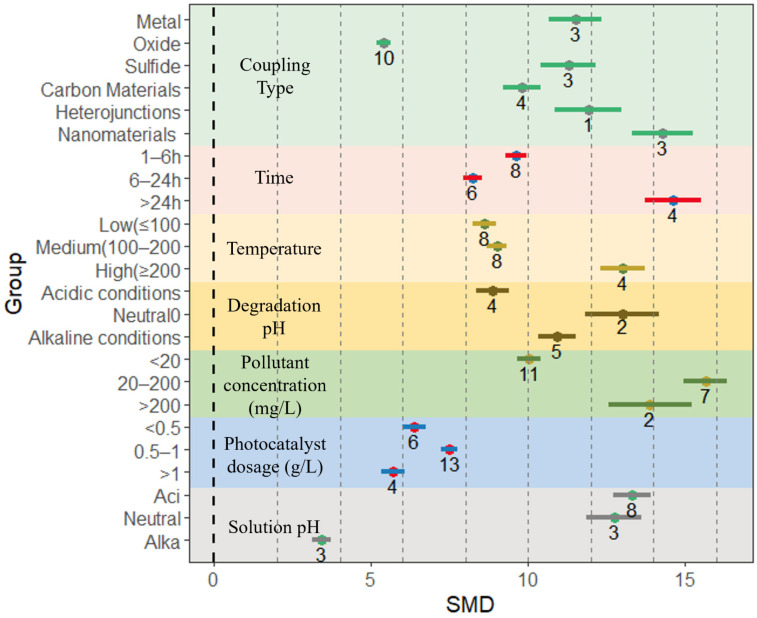
The degradation efficiency for organic pollutants of BiVO_4_-based composites is mainly influenced by the type of coupling, degradation time, degradation temperature, degradation pH, pollutant concentration, photocatalyst dosage, and solution pH. Horizontal error bars indicate 95% confidence intervals. (Colored lines indicate effect size intervals and numbers indicate the amount of literature for that component).

**Table 1 nanomaterials-13-02352-t001:** Summary of information related to BiVO_4_-based composites in the selected literature.

Appearance	Preparation Method	Degradation Efficiency of Single BiVO_4_ (%)	Degradation Efficiency after Loading (%)	Cycle Performance (%)/Cycle Count	Advantages	References
Microsphere	Photo-deposition method	12.1	90.5	-	High efficiency; potential applications in solar energy, water decomposition, and medicine	[47]
N-decahedron	Hydrothermal method	31.7	99.0	99.0/5 times	Good photocatalytic stability; significant improvement of photocatalytic performance	[54]
3D z-shape	Hydrothermal method	45	100	≥85/5 times	Novel and efficient in removing both heavy metals and organic pollutants from water	[55]
Nanorod	Hydrothermal and co-precipitation methods	34	87	77.5/5 times	Good structure and compound performance; can effectively degrade organic pollutants such as RhB and TC; good stability	[50]
Peanut-like nanorods with monoclinic structure	Hydrothermal method	45	96	92/5 times	Low cost and high photocatalytic activity; can be loaded on clay to degrade organic pollutants	[56]
Nanosheet	Hydrothermal and photoreduction methods	58.6	96.2	-	Simultaneous removal of Cr(VI) and CIP pollutants; high photocatalytic activity and stability for degradation	[57]
Nanoparticle aggregation	Hydrothermal method	56	96.23	-	Excellent size and structural homogeneity; good photocatalytic activity for visible-light-driven photocatalysts for water remediation	[58]
Band	Hydrothermal method	56	98	87/4 times	-	[59]
Nano-floral	Hydrolysis method	45.94	90.20	88.58/4 times	Large specific surface area and high photocatalytic efficiency	[60]
Ribbon	Simple solvent heat path	-	-	-	Higher surface area and crystallinity; high activity; recyclable	[61]
Sheet	Ultrasound-assisted solvent heat method	67	99.98	94/5 times	-	[62]
Rod-shaped pellet	Ultrasonic-assisted method	28	100	97/5 times	Large surface area; suitable band gap and small crystal size; excellent visible light photocatalytic activity	[63]
Sheet	Hydrothermal method	43	85	-	-	[48]
Sheet	Ultrasonication	49	92	95/5 times	Good contact surface structure; high photocatalytic activity; strong optical absorption ability; good adsorption for organic pollutants; low level of electron–hole pair compounding	[64]
Nanosheet aggregation	Hydrothermal method	36	86.7	-	High reusability and photocatalytic performance	[65]
Lump	Ultrasonic-assisted method	47	93.6	78.7/6 times	Large specific surface area; high number of increased active sites; good visible light absorption range and construction of heterojunctions; high stability	[66]
Nanowires or nanorods	Hydrothermal method	48	92	-	High oxidation and reduction capacity; economical and effective	[67]
Spherical nanoparticles	Liquid precipitation mechanical mixing method	14.57	90.14	78/3 times	Large specific surface area; high photocatalytic activity	[68]
Olivine nanoparticles	One-step solvent heat method	34	99	-	Synthesis strategy is simple, easy, environmentally friendly, and scalable	[69]
Regular octahedron	In situ simple precipitation method	47.50	93.67	89.29/5 times	Simultaneous removal of Cr(VI) and MB-mixed contamination, with good reusability and stability	[70]
Dumbbell-shaped	Visible-light-assisted photocatalysis	85	99	-	BVO/rGO has small particle size, strong adsorption capacity and high photocatalytic activity	[71]
Hexagonal nanorods	Chemical precipitation method	-	-	91.35/5 times	Strong adsorption capacity; good photocatalytic activity	[72]
Porous tubular	-	58.60	97.66	80/5 times	High separation rate of photogenerated charges and high efficiency of catalytic degradation of organic pollutants	[73]
Flowery sphere	In situ technology	26	60	32/9 times	Photogenerated carriers can be effectively separated with high oxidation and reduction capacities	[74]
Sheet	Easy precipitation method	38	94	-	Excellent performance in degrading organic dyes and good reusability	[75]
Spherical cluster	Two-step solvent heat method	77	89	85.3/5 times	Excellent adsorption performance and abundant active sites	[76]
Irregular spherical/oval clusters	Hydrothermal method	71	97	-	Low compounding rate of electron–hole pairs; small energy band gap; strong electron capture ability; high photocatalytic activity	[77]

**Table 2 nanomaterials-13-02352-t002:** Effects of different preparation and degradation conditions on the photocatalytic properties of BiVO_4_-based composites.

	Variable	Group	Sample Size (n)	95% CI (SMD)		Effect Size (%)
				Lower	Upper	
Preparation conditions	Coupling type	Metal	3	7.49	8.69	134.79
		Oxide	10	3.94	4.73	62.75
		Sulfide	3	8	9.36	152.19
		Carbon materials	4	7.02	7.96	79.31
		Heterojunctions	2	12.73	15.13	136.40
		Nanomaterials	4	12.96	14.56	90.10
	Time	1–6 h	9	6.72	7.2	91.90
		6–24 h	7	8.87	9.56	59.59
		>24 h	4	14.26	16.12	145.34
	Temperature	Low (≤100)	9	4.65	5.12	83.37
		Medium (100~200)	9	8.84	9.46	77.69
		High (≥200)	4	11.31	12.62	121.33
	Degradation pH	Acidic conditions	4	8.33	9.41	88.08
		Neutral	4	10.92	12.36	125.47
		Alkaline conditions	5	10.33	11.52	72.56
Degradation condition	Pollution concentration (mg/L)	<20	13	6.84	7.33	93.48
		20–200	7	10.66	11.69	105.83
		>200	2	7.26	12.92	285.03
	Photocatalyst dosage (g/L)	<0.5	6	5.28	5.97	95.45
		0.5–1	14	7.00	7.52	112.02
		>1	4	4.56	5.21	65.32
	Solution pH	Acidic	8	9.02	9.91	119.93
		Neutral	3	13.79	15.82	115.74
		Alkaline	4	4.02	5.19	52.82

## Data Availability

Not applicable.

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
