# Peer review of "A Meta-Analysis of Influencing Factors on the Activity of BiVO4-Based Photocatalysts"

_nanomaterials, 2023, doi:10.3390/nano13162352_

Round 1
Reviewer 1 Report
The comments are in the attached file.

Reviewer 2 Report
This work gives a basic detail on different influencing factors on the photocatalytic process of BiVO4 and their composites. This manuscript can be accepted, but some questions should be addressed.
1. The authors failed to elucidate the BiVO4 vs composites photocatalytic activity.
2. In table1, the detail of the information is not enough for the readers. It should be clear with more information instead of a simple summary.
3. Why does the formation of heterojunction increase photocatalytic activity (Section 4.3.3)? The discussion looks common, and it should be in scientific aspects.
4. From this paper, the necessity of developing BiVO4- based catalyst was not emphasized, the authors could add the related discussion.
5. Line 125, what is the meaning of Cr(VI) in categories?
6. Line 132-133, how authors detected the loading dosage of catalysts? Is it correct on the discussion?
7. Line 244, the photodegradation is only consider for reduction?
8. Line 379, why figure 20 was included?
9. The effect of the degradation pH is not well understood, so needs to be made clear.
10. The conclusion and outlook have little contribution to the review topic. A clear catalogue of this article should be added for better readability.
Minor editing of English language required.
Reviewer 3 Report
This paper focuses on the influence of BiVO4-based photocatalyst preparation and pollutant degradation conditions on photocatalyst activity. It is important to analyze previous literature to optimize the synthesis method and conditions of use of photocatalysts. However, finding them in this paper takes work, and a more detailed discussion is needed. For example, there are many kinds of metals and oxides, and each has a different effect. For these reasons, I cannot recommend the publication of this manuscript in its present form.
Round 2
Reviewer 2 Report
This manuscript is ready for the publication.
Reviewer 3 Report
The authors have revised properly the manuscript based on the reviewer’s comments. I think that the manuscript in the present form could be accepted for publishing.